# Fast Benchmarking of Accuracy vs. Training Time with Cyclic Learning Rates

**Jacob Portes**[*]
Columbia University, MosaicML
jacob@mosaicml.com

**Davis Blalock**
MosaicML
davis@mosaicml.com

**Cory Stephenson**
MosaicML
cory@mosaicml.com

**Jonathan Frankle**
MosaicML
jonathan@mosaicml.com

## Abstract

Benchmarking the tradeoff between neural network accuracy and training time is computationally expensive. Here we show how a multiplicative cyclic learning rate schedule can be used to construct a tradeoff curve in a single training run. We generate cyclic tradeoff curves for combinations of training methods such as Blurpool, Channels Last, Label Smoothing and MixUp, and highlight how these cyclic tradeoff curves can be used to efficiently evaluate the effects of algorithmic choices on network training.

In order to make meaningful improvements in neural network training efficiency, ML practitioners must be able to compare between different choices of network architectures, hyperparameters, and training algorithms. One straightforward way to do this is to characterize the tradeoff between accuracy and training time with a "tradeoff curve." Tradeoff curves can be generated by varying the length of training for each model configuration; longer training runs take more time but tend to reach higher quality (Figure 1C). For a fixed model and task configuration, this method of generating tradeoff curves is an estimate of the theoretical **Pareto frontier**, i.e. the set of all of the best possible tradeoffs between training time and accuracy, where any further attempt to improve one of these metrics worsens the other.[2]

Accuracy-time tradeoff curves also allow us to make quantitative statements about different hyperparameter choices and training algorithms [Leavitt, 2022, Blakeney et al., 2022]. For example, training the same network with Blurpool [Zhang, 2019], Channels Last,[3] Label Smoothing [Szegedy et al., 2016, Müller et al., 2019], and MixUp [Zhang et al., 2017] shifts the tradeoff curve left along the time axis and up along the accuracy axis, leading to a Pareto improvement in both accuracy and speedup relative to the baseline (Figure 1C). Our main research question in this work is: *Given a fixed training configuration (e.g. architecture, objective function, and regularization methods), how can we efficiently generate an accuracy vs. training time tradeoff curve?*

One approach might be to do multiple training runs with different training durations and construct a tradeoff curve from the final accuracies. For example, we could train a network for three different durations of 64, 128, and 256 epochs, and fit a parameterized function to interpolate the rest of the tradeoff curve. In this scenario, training would take 448 epochs total. For the remainder of this paper,

---

[*]Corresponding author. Work done during a research internship at MosaicML. Code available at https://github.com/jacobfulano/cyclic-learning-rate-schedules

[2]We choose to use the language of "tradeoff curve" instead of Pareto frontier with the understanding that the tradeoff curve is likely a lower bound of the true Pareto frontier.

[3]We refer here to the Channels Last memory format in PyTorch https://pytorch.org/blog/tensor-memory-format-matters/

Has it Trained Yet? Workshop at the Conference on Neural Information Processing Systems (NeurIPS 2022).

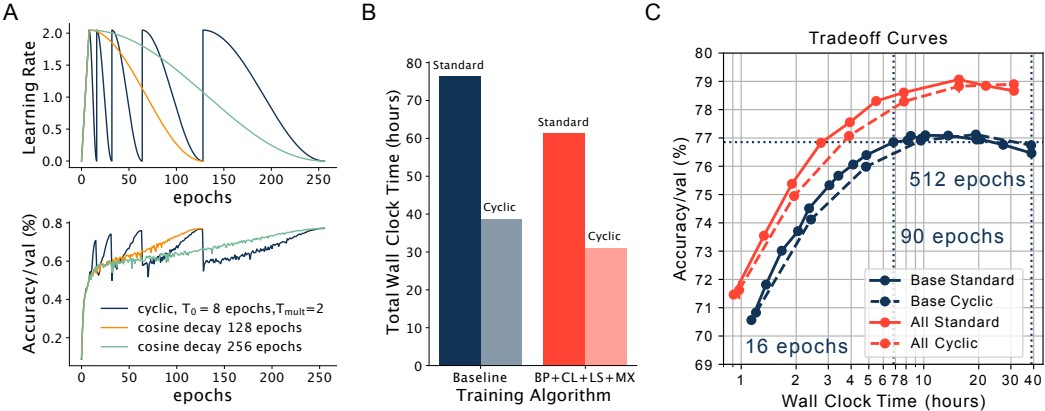

Figure 1: Accuracy-time tradeoff curves can be efficiently constructed with cyclic learning rate schedules. (A) Multiplicative cyclic learning rate schedules increase and decrease the learning rate with increasing periods (*top*), while cosine decay learning rate schedules decrease monotonically after linear warm up. Cyclic learning rate schedules lead to cycles in accuracy during training (*bottom*). (B) Total wall clock time for generating a tradeoff curve with 6 sample points at 16, 32, 64, 128, 256 and 512 epochs. Training 6 separate networks by scaling the cosine decay schedule (Standard, *dark blue* and *crimson*) takes twice as long as training a single network for 512 epochs with a cyclic learning rate schedule (Cyclic, *light blue and crimson*). Total wall clock time is shown for baseline training (*blue*) and training with Blurpool (BP), Channels Last (CL), Label Smoothing (LS), and MixUp (MX). (C) Cyclic peaks in accuracy for cyclic learning rate schedules can be used to generate accuracy vs. time tradeoff curves. This is shown for baseline training of ResNet-50 on ImageNet (*blue, dashed line*) and for training ResNet-50 with BP, CL, LS, and MX, (*crimson, dashed line*).

we refer to any tradeoff curve generated this way as a **standard tradeoff curve**. Another approach might be to do one long training run (e.g. for 256 epochs) and sample points on the tradeoff curve during training. However, modern training best practices involve gradually decaying the learning rate over the course of training. Many of the points sampled in this procedure will therefore be at intermediate states where training is not complete and the learning rate schedule is atypically high, leading to lower-than-normal model quality (this can be seen in Figure 1A). Neither of these approaches is satisfactory. The first approach is too computationally expensive, and the second produces low-fidelity results that are not indicative of standard performance.

Cyclic learning rate schedules were first proposed in Smith [2017], and essentially increase and decrease the learning rate periodically during training. One version of a cyclic schedule consists of repeated cosine decay with periods that increase multiplicatively, which we call **multiplicative cyclic learning rate schedules** (Figure 1A).[4] We suspected that cyclic learning rate schedules could be used to efficiently construct accuracy-time tradeoff curves, by storing validation accuracy peaks that track cyclic learning rate troughs (Figure 1C).

In this paper, we show that multiplicative cyclic learning rate schedules can be used to efficiently construct accuracy-time tradeoff curves for benchmarking on ImageNet. In particular, we show that multiplicative cyclic learning rate schedules can be used to construct an accuracy-time tradeoff curve in a single training run, saving time and money. We then explore how these curves change for regularization and speedup methods such as Blurpool, Channels Last, Label Smoothing, and MixUp, both individually and in combination. Although using this method necessarily changes the outcome of training, we still find that it provides similar-fidelity information about the relative efficacy of different training methods. This is particularly useful for characterizing the accuracy-time tradeoff more efficiently, and for making meaningful comparisons between choices of architectures, hyperparameters and network training methods.

**Related Work** While Smith [2017] proposed various sawtooth-like schedules, Loshchilov and Hutter [2016] showed that a schedule designed with cycles of a cosine decay followed by a restart led to

---

[4]In the original paper, the schedule is called SGDR, i.e. Stochastic Gradient Descent with Restarts. In PyTorch, the scheduler is called `CosineAnnealingWarmRestarts`

competitive accuracy on CIFAR-100. In practice, cyclic learning rate schedules have been used for ensembling methods [Huang et al., 2017] and Stochastic Weight Averaging [Izmailov et al., 2018]; they have also been the subject of theoretical work [Goujaud et al., 2022]. The use of Pareto frontiers in neural network research is surprisingly limited, and is mostly restricted to the subfield of multi-objective optimization (MOO) for multitask learning; recent examples include Lin et al. [2020] and Navon et al. [2020]. We extend this framework to evaluate the effects of algorithmic choices on network training efficiency.

**Results** We trained ResNet-50 on ImageNet with (noncyclic) cosine decay learning rate schedules for various training durations, and cyclic learning rate schedules for 512 epochs. All runs began with a linear warm-up over 8 epochs to the maximum learning rate, and all learning rate schedules were stepwise and not epoch-wise. For standard (noncyclic) training, runs were done across 3 seeds for each data point at 16, 32, 64, 128, 256, and 512 epochs. Cyclic runs were done across 3 seeds. All experiments report top-1 validation accuracy.

**Multiplicative Cyclic Learning Rate Schedules Efficiently Estimate Tradeoff Curves** How efficient might a cyclic learning rate schedule be? The cyclic learning rate schedule in Loshchilov and Hutter [2016] is defined as $\eta_t = \eta_{min} + \frac{1}{2}(\eta_{max} - \eta_{min})(1 + \cos(\frac{T_{cur}}{T_i}\pi))$, where $\eta_t$ is the learning rate at epoch $t$, $\eta_{max}$ and $\eta_{min}$ are the maximum and minimum learning rates, period $T_i$ is the number of epochs between two cycles, and $T_{cur}$ is the number of epochs since the end of the previous cycle. For a multiplicative cyclic learning rate schedule, the period for each cycle increases such that $T_i = rT_{i-1}$, where $r$ is the multiplicative factor and $i$ denotes the cycle.

The ratio of the time to construct a standard tradeoff curve versus the time to construct the cyclic tradeoff curve, where both tradeoff curves sample the same exact points, can be expressed in the following formula:

$$\text{speedup ratio} = \frac{\sum_{i=0}^{n-1} T_0 r^i}{T_0 r^{n-1}} = \frac{1 - r^n}{(1-r)r^{n-1}} \tag{1}$$

where $T_0$ is the starting number of epochs, $r$ is the multiplicative factor, and $n - 1$ is the total number of sampled points (and number of cycles). When $r = 2$, this ratio is approximately 2, meaning that using a cyclic learning rate schedule should in principle lead to $\sim 2\times$ speedup (Figure 1B, Figure 3D).

We used multiplicative cyclic learning rate schedules with doubling periods, and found that they approximated the baseline standard tradeoff curve quite well. For the rest of this work, we refer to these *multiplicative* curves as **cyclic tradeoff curves**. We designed the learning rate schedule to linearly warm up for 8 epochs to a maximum value of 2.048, and then cycle with a cosine decay for increasing periods of 8, 16, 32, 64, 128 and 256 epochs, for a total of 512 epochs of training. We then stored maximum accuracy values at the end of each cycle, and plotted an accuracy-time tradeoff curve. This cyclic tradeoff curve matched our ResNet-50 ImageNet baseline standard tradeoff curve (with a difference of $-0.084\%$ on average across three seeds for all points, Figure 1C, Figure 2). This result held across choices of starting periods (e.g. starting with $T_0 = 5$ epochs, then doubling to $T_1 = 10$, $T_2 = 20$ etc. epochs; data not shown). We found that cyclic learning rate schedules with constant periods were not able to approximate tradeoff curves (Appendix B, Figure S1).

**Cyclic Learning Rate Tradeoff Curves are Predictive Across Training Methods** There are a handful of demonstrated methods that modify the training procedure to achieve better tradeoffs between the final model accuracy and the time to train the model. Some of these methods include Blurpool [Zhang, 2019], Channels Last, Label Smoothing [Szegedy et al., 2016, Müller et al., 2019], and MixUp [Zhang et al., 2017]. In our benchmarking work, we wanted to find methods in the literature that led to clear Pareto improvements. We then asked whether multiplicative cyclic learning rate schedules could be used to construct competitive accuracy-time tradeoff curves for separate methods such as Blurpool (BP), Channels Last (CL), Label Smoothing (LS), and MixUp (MX). We found that using multiplicative cyclic learning rate schedules allowed us to generate tradeoff curves of comparable quality to standard tradeoff curves, but at substantial time savings ($\sim 2\times$).

Cyclic learning rate tradeoff curves were able to match baseline standard tradeoff curves for BP and CL almost exactly (BP exceeded by $0.07\%$, and CL by $0.09\%$, on average across all points, Figures 2A-B). Note that the simple CL modification on RTX 3080 GPUs led to a leftward shift of the tradeoff curve along the time axis (*solid cyan line*); thus training a network for 512 epochs took approximately 30 hours instead of 40 hours. BP also led to strong improvements for shorter training

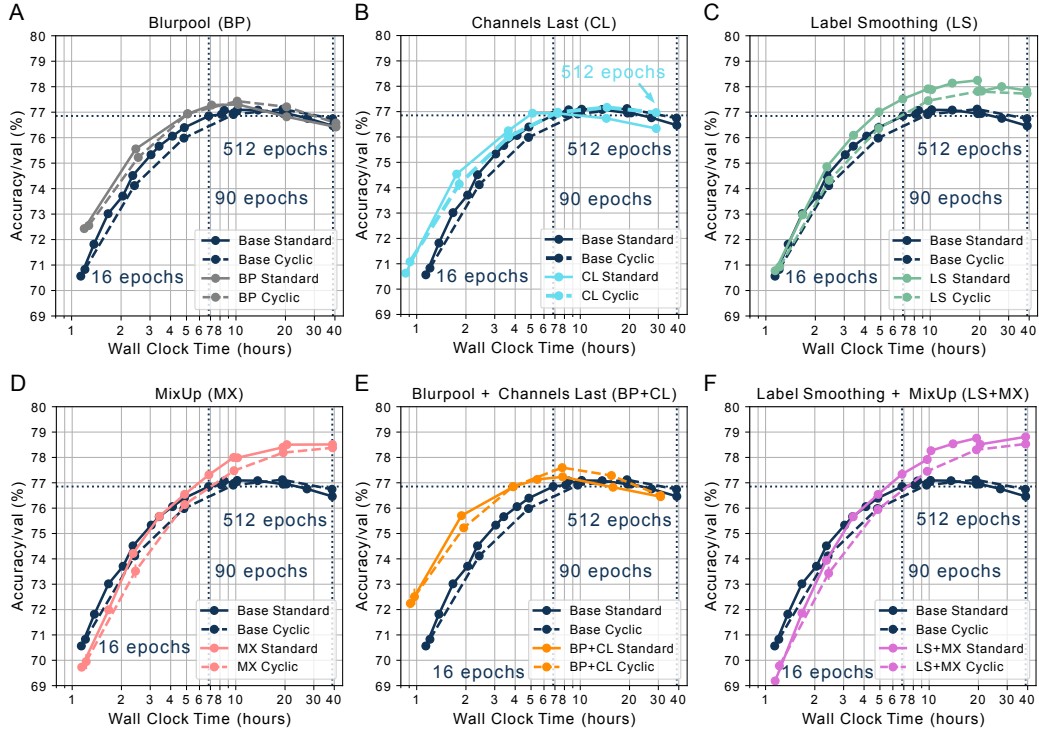

Figure 2: Cyclic learning rate tradeoff curves are similar to standard tradeoff curves across training methods (A) Blurpool (BP) tradeoff curve constructed with the standard method (*solid gray*) and cyclic approach (*dashed gray*) are similar. (B) The simple Channels Last (CL) modification on 3080 GPUs leads to a leftward shift along the time axis. (C) Label Smoothing (LS) leads to improvements later in training (*green*). (D) MixUp (MX) leads to accuracy improvements later in training, and prevents overfitting (*pink*). (E) Tradeoff curve for networks trained with both Blurpool and Channels Last (BP+CL) (*gold*). (F) Tradeoff curves for Label Smoothing and MixUp (LS+MX) (*magenta*).

times (e.g. 16 epochs). Cyclic learning rate tradeoff curves were slightly below baseline standard tradeoff curves for LS and MX, on average differing roughly by $-0.44\%$ for LS and $-0.39\%$ for MX across all points (Figures 2C-D). Importantly, cyclic tradeoff curves for these two cases still trended with the standard tradeoff curves.

Constructing a separate accuracy-time tradeoff curve for every training method and/or combination of methods (for example LS + MX) is also combinatorially expensive. We therefore asked whether cyclic learning rate schedules could accurately construct tradeoff curves for combinations of training methods. We trained ResNet-50 with both BP and CL (BP + CL, Figure 2E), and found that the cyclic learning rate schedule tradeoff curve matched the standard tradeoff curve (exceeded by $0.08\%$, on average across all points).

We then trained ResNet-50 with the combined methods of LS and MX (LS + MX, Figure 2F), and found that cyclic tradeoff curves trended with standard tradeoff curves, but were slightly lower (by $-0.46\%$ on average). Finally, we trained ResNet-50 with the combined methods of BP, CL, LS and MX (BL+CL+LS+MX, Figure 1C), and found that cyclic tradeoff curves trended with the standard tradeoff curves (*crimson dashed line*, lower by $-0.24\%$ on average across all points).

When comparing training methods such as LS and MX, can we really trust that improvements in the Pareto frontier over the baseline will be matched by a tradeoff curve constructed with a cyclic learning rate? One way to assess this is simply to compare the relative improvements over standard or cyclic baseline for each method. We found that relative improvements held between standard and cyclic tradeoff curves (Figure 3A, Figure S2).

For example, when the baseline standard tradeoff curve is subtracted from the Blurpool standard tradeoff curve, it is clear that Blurpool provides a $2\%$ accuracy boost early on in training, but provides

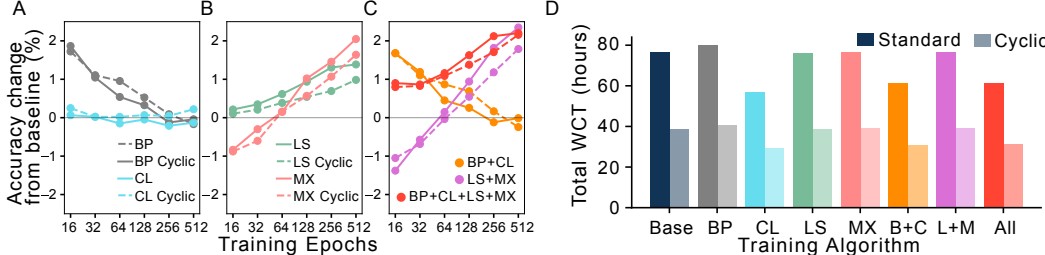

Figure 3: Relative improvements hold between standard and cyclic tradeoff curves (A) Relative improvements over baseline for each method can be plotted by subtracting the baseline standard/cyclic tradeoff curve from the standard/cyclic tradeoff curve of a particular method. Relative improvements over baseline for standard tradeoff curves (*solid lines*) are captured by cyclic tradeoff curves (*dashed lines*). (A) Blurpool and Channels Last (B) Label Smoothing and MixUp (C) Combinations of methods, including Blurpool + Channels Last (*gold*), Label Smoothing + Mixup (*magenta*), and all four methods combined (*crimson*) (D) Total wall clock time (WCT) for standard tradeoff curves with six sampled points at 16, 32, 64, 128, 256 and 512 epochs is almost twice as large as cyclic tradeoff curves constructed with the same six sampled points.

no accuracy boost after 256 epochs (Figure 3A, *gray solid line*). Subtracting the baseline cyclic tradeoff curve from the Blurpool cyclic tradeoff curve shows the same trend (Figure 3A, *gray dashed line*).

We then plotted the total wall clock time for standard tradeoff curves and cyclic tradeoff curves with six sampled points at 16, 32, 64, 128, 256 and 512 epochs (Figure 3D). As expected, standard tradeoff total wall clock time was almost twice as long as cyclic tradeoff curve total wall clock time across all training methods and combinations of methods (see Equation 1).

**Discussion** We have shown here that multiplicative cyclic learning rate schedules can be used to construct an accuracy-time tradeoff curve in a single training run, saving time and money. This is a practical means of assessing changes in tradeoff curves caused by different architectures, hyperparameters, and training methods such as Label Smoothing and MixUp. We hope that this is a small step towards framing training choices in terms of tradeoff curves and Pareto improvements.

It is somewhat surprising that multiplicative cyclic learning rates are able to approximate Pareto frontiers when constant period cyclic learning rates do not (Figure S1). Why does this work, particularly for increasing periods? One possibility might have to do with different phases of learning [Golatkar et al., 2019, Frankle et al., 2020]. For example, rapid changes in learning rate might help find better minima early in training (10-20 epochs), while very slow changes in learning rate might help find minima in already "good" basins late in training (>90 epochs). As training continues, it might become more difficult to find better minima, implying that it might be beneficial to take longer for each new cycle.

Cyclic learning rate schedules were popularized by Fast.ai [Howard and Gugger, 2020], but are not widely used in the research community. This is likely due to the fact that, with current best practices on benchmarks like ImageNet, cyclic learning rate schedules do not necessarily lead to improved accuracy when compared to cosine decay schedules.

In many instances, the cyclic tradeoff curve underestimated the standard tradeoff curve by a margin of 0.5% validation accuracy. There are many possible variations to our approach that might improve this gap. One idea is to allow the entire cyclic learning rate schedule to decay over the course of learning; this is one of many schedule variations explored by the literature [Schmidt et al., 2021]. Another idea would be to apply a cyclic learning rate schedule to other hyperparameters such as weight decay, batch size, and gradient clipping (in line with Smith [2022]).

Our results primarily hold for convolutional networks such as ResNet-50 and ResNet-101 (data not shown) applied to computer vision tasks such as CIFAR-10 (data not shown) and ImageNet. Can a similar approach be applied to other architectures such as transformers (that take significantly longer to train than ResNets) and to other domains such as NLP? We hope to explore this in future work.

**Acknowledgements** An earlier version of this work appeared as a blog post `https://www.mosaicml.com/blog/efficiently-estimating-pareto-frontiers`. Thanks to Abhinav Venigala, Matthew Leavitt, and Kobie Crawford for fruitful discussions.

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

# A   Experimental Details

We trained ResNet-50 on ImageNet with noncyclic and cyclic learning rate schedules for various training durations. All ImageNet experiments used 8 RTX 3080s with mixed precision training, 8 dataloader workers per GPU, a batch size of 2048, and a gradient accumulation of 4. We used SGDW [Loshchilov and Hutter, 2017] with a maximum learning rate of 2.048, momentum of 0.875, and weight decay 5e-4 (these values were found to be optimal in a hyperparameter sweep; the learning rate of 2.048 was chosen to be proportional to the batch size). All runs began with a linear warm-up over 8 epochs to the maximum learning rate, and all learning rate schedules were stepwise and not epoch-wise. For standard (noncyclic) training, runs were done across 3 seeds for each data point at 16, 32, 64, 128, 256, and 512 epochs, and for 1 seed for other data points. Cyclic runs were done across 3 seeds. Experiments using cosine decay learning rate schedules were run using the `CosineAnnealingLR` scheduler in PyTorch, while cyclic learning rate schedules were implemented using the PyTorch `CosineAnnealingWarmRestarts` scheduler. Training methods such as Blurpool Zhang [2019], Channels Last, Label Smoothing Szegedy et al. [2016], Müller et al. [2019] and MixUp Zhang et al. [2017] were implemented using the MosaicML Composer library for PyTorch `https://github.com/mosaicml/composer`.

# B   Cyclic Learning Rate Schedules with Constant Periods

We used cyclic learning rate schedules with **constant** periods to construct accuracy-time tradeoff curves, and examined the effect of varying the periods ($T = 10, 20, 30, 40$, and $50$ epochs). We found that cyclic learning rate schedules with shorter periods led to suboptimal tradeoff curves, while cyclic learning rates with longer (constant) periods led to curves that were on the standard tradeoff curve but were not well sampled (Figure S1A-B).

Historically, there was a flurry of work exploring alternative learning rate schedules from exponential decays to trapezoids; cyclic learning rates fall on the more exotic end of the spectrum. We found mixed results for other cyclic schedules such as sawtooth/triangle schedules as in Smith [2017]. For a review of different schedule shapes, see Table 3 of Appendix A in Schmidt et al. [2021] as well as Wu et al. [2019].

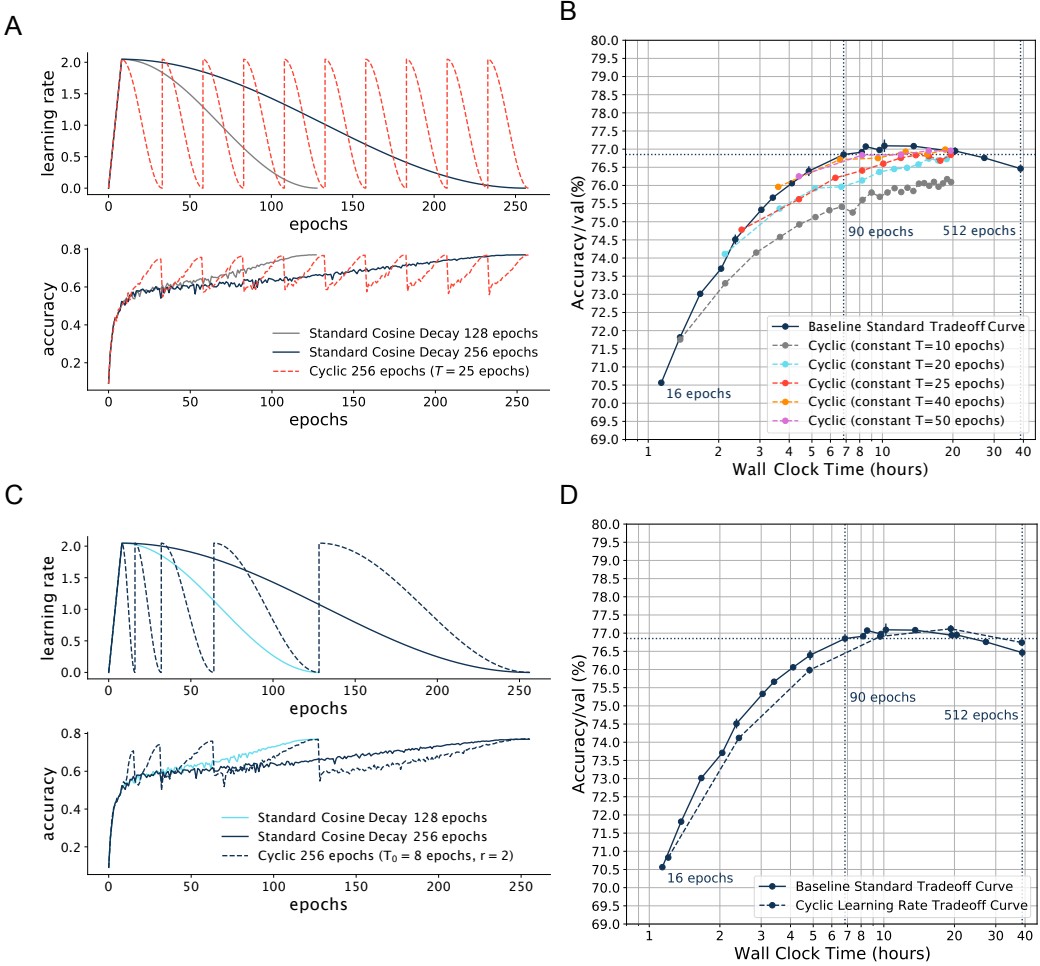

Figure S1: Constructing accuracy-training time tradeoff curves with constant and multiplicative cyclic learning rate schedules (A) Cyclic learning rate schedule with a constant period of 25 epochs (*top*) leads to a validation accuracy that is also cyclic (*bottom*). (B) Accuracy-training time tradeoff curves can be constructed with cyclic learning rate schedules of different constant periods, for example 10 (*gray*), 20 (*cyan*), 25 (*crimson*), 40 (*gold*) and 50 epochs (*magenta*). (C) Cyclic learning rate schedule with multiplicative periods of 16, 32, 64 and 128 epochs (*top*) also lead to a validation accuracy that is cyclic (*bottom*). (D) An accuracy-time tradeoff curve can be constructed with a multiplicative cyclic learning rate schedule (*dashed line*). This tradeoff curve closely resembles the standard tradeoff curve constructed from individual training runs with scaled cosine decay schedules (*solid line*).

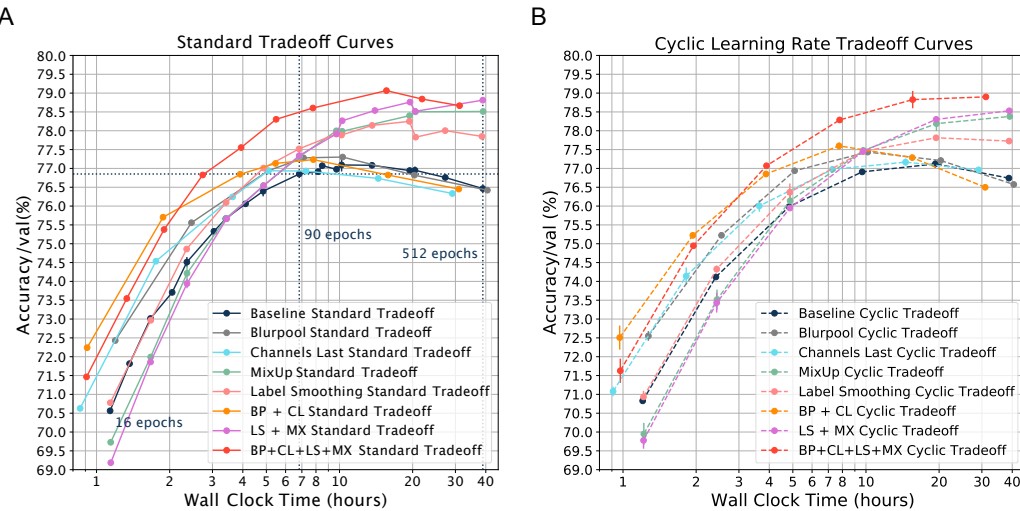

Figure S2: Relative ordering of tradeoff curves between training methods is preserved. (A) Standard tradeoff curves across training techniques. (B) Accuracy vs. training time tradeoff curves using multiplicative cyclic learning rate schedules across training methods. Same data as in Figures 1 and 2.

