# OpenReview forum: "Fast Benchmarking of Accuracy vs. Training Time with Cyclic Learning Rates"
_NeurIPS.cc/2022/Workshop/HITY — HITY Workshop NeurIPS 2022_

### Official Review · Reviewer_Tt7c · 2022-10-05
**Neat idea to use multiplicative cyclic learning rate schedules to construct an accuracy-time tradeoff curve in a single run**

**Rating:** 1
**Confidence:** 2

**Review:**

This paper proposes to use the peaks and droughts in training with multiplicative cyclic learning rate schedules, to construct approximations to accuracy-training tradeoff curves. While this is an ad hoc approximation, it has the advantage to be done in one training pass. The idea sounds crazy to me, but the experiments are actually pretty encouraging in the sense that the approximations are pretty accurate. Hence, I vote to accept this paper.

---

### Official Review · Reviewer_sNcK · 2022-10-06
**Well written paper**

**Rating:** 1
**Confidence:** 4

**Review:**

This paper shows that multiplicative cyclic learning rate schedules (cosine decay with warm restarts) can be used to construct an accuracy-time tradeoff curve in a single training run for a ResNet-50 trained on Imagenet.

The paper is clearly written and understandable throughout. The topic fits well with this workshop.

Drawbacks are:
- Only data for a ResNet-50 on Imagenet is presented. It is unclear if this approach will work across datasets, loss functions, opimizers, architectures, domains... .
- The authors claim that their approach also works on ResNet-50 and ResNet-101 trained on CIFAR-10, but are not providing any data. Please provide this data.
- The source code is not provided, thus it is impossible to evaluate if the experiments are implemented correctly.

---

### Official Review · Reviewer_csBR · 2022-10-10
**Using Cyclical Learning Rates for Benchmarking**

**Rating:** 1
**Confidence:** 4

**Review:**

The paper introduces the idea of using cyclical learning rate schedules to benchmark the performance of different algorithmic and model changes. Usually, benchmarking these methods requires training for multiple different budgets (e.g. for 16, 32, 64, ... epochs), but cyclical learning rates allow an approximation of these results with a single training run (for the full duration).

The paper introduces a neat and practically helpful method for faster benchmarking of algorithmic and/or model changes. The experiments show that the cyclical training can consistently provide a decent approximation of the "true" tradeoff curve while requiring only half of the training costs.

Feedback:
- Line 67 mentions that the experiments report top-1 validation accuracy. Judging from the results, I would guess that this describes the `final` achieved performance (i.e. what the final iterate produces) and not the `best` achieved performance (i.e. what the `best` parameters observed during training achieve), right? Otherwise, the tradeoff curves should never be decreasing (for the cyclical learning rate). However, in Line 73 you describe "we then stored maximum accuracy values at the end of each cycle". I don't understand what set the `maximum` refers to. I would be interested in seeing the results using the `best` performance instead of the `final` (would be sufficient to show it in the appendix). This would not require any further experiments.
- Could you provide some more details on how you set the (upper) learning rate of both the standard and the cyclical training? Line 72 mentions using a value of 2.048 for the cyclical training. Could it be that some methods just work better with this learning rate (without providing better performance at a tuned value)?
- I would be interested in hearing your thoughts on whether this type of benchmarking "trick" could also be used in other domains, besides ImageNet training. Would you anticipate any problems using the cyclical learning rate schedule in other domains?

Nits:
- Line 56: Should probably use a \citep, e.g. "[Huang et al., 2017]". This occurs multiple times.
- Line 60: doubling of "to".

---

### Decision · Program_Chairs · 2022-10-20

Accept